# A Study on the Corrosion Resistance of a Coating Prepared by Electrical Explosion of 321 Metal Wire

**Ye Liu** [ID]**, Qiuzhi Song *** [ID]**, Hongbin Deng, Yali Liu** [ID]**, Pengwan Chen and Kun Huang** [ID]

College of Mechanical and Electrical Engineering, Beijing Institute of Technology, Beijing 100081, China; ly1989bit@163.com (Y.L.); denghongbin@bit.edu.cn (H.D.); pwchen@bit.edu.cn (P.C.); hkhk1008@163.com (K.H.)
* Correspondence: qzhsong@bit.edu.cn; Tel.: +86-10-68912044

**Abstract:** Corrosion is known as a breakdown effect that causes the deterioration of substances in enriched petroleum/gas conditions. This reaction occurs in all materials, which is highlighted in alloys. In the present study, the morphological properties, as well as the corrosion resistance behavior of the AISI1045 steel substrate coated with 321 austenitic stainless steel metal particulate fillers, were investigated. The electro-explosive spraying technique was employed to achieve a homogenous coating on the substrate surface. According to the results, the grain size of the 321 austenitic stainless steel coating layer was shrunk and reduced to 1–3 μm after the coating procedure. The coated layer also showed a homogenous and uniform thickness with an average value of 137 μm. Also, the average adhesion strength of 49.21 MPa was obtained between the sprayed coating and the substrate. The analytical analysis found the presence of Fe-Cr and Fe-Ni phases in the coating layer. The hardness of the original metal wire is 186 HV, and the microhardness of the coating after spraying is 232 HV. After subjecting the specimen to the corrosion examination, a 0.1961 mm/a corrosion rate was obtained for up to 120 h. Moreover, the corrosion products of $CaCO_3$, $Fe_3O_4$, and $MgFe_2O_4$ were determined by XRD analysis. Furthermore, the observed results were further confirmed by the data obtained from EPMA and EDS evaluations. Hence, this study implies the beneficial role of electro-explosive sprayed alloy 321 austenitic stainless steel in creating a protective layer against corrosion on 45 steel substrate in an enriched oil/water environment.

**Keywords:** electro-explosive spraying; alloy 321; coating corrosion; corrosion products; corrosion kinetic curve

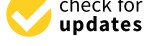



## 1. Introduction

In recent decades, the increasing demand for energy consumption has caused a rise in the need for mining processes and transition industries [1]. Metal-based equipment is a key building block in the highlighted industries. In this era, it is highly recommended to optimize and modify the performance of metal-based equipment toward monitoring energy costs, as well as sustaining the environment. Steel is a widely used metal in such industries, faced with corrosion and thus degradation challenges when it is exposed to oil, water, and atmosphere [2,3]. In aqueous conditions, the negative ions present in the water mainly affect steel corrosion. As an example, $Cl^-$ is known as a corrosive anion during the mining process, causing acceleration of the corrosion rate, failing of the pipeline perforation, and irreparable economic losses. Therefore, the common steel pipelines cannot be employed in a mining environment without modifications [4,5].

Currently, a coating is recognized as the most feasible, cost-efficient, and efficient approach toward increasing corrosion resistance, divided by two main non-metallic and metal-based techniques [6,7]. For example, the epoxy coating prepared by Tong Liu et al. has efficient self-healing performance [8]. Sepehr Yazdani et al. optimized the corrosion performance of Ni-B diamond coatings using noise theory [9]. Compared to metal-based strategies, non-metallic coatings have shown poor resistance in high temperatures, leading

to subsequent disadvantages for the environment. Generally, a metal-based coating could be conducted through galvanizing, sherardizing, electroplating, and thermal spraying methods [10–12]. Salimeh Hasanbeigi et al. optimized the corrosion performance of Mg Al coatings by improving parameters [13]. Sangeetha et al. studied the corrosion performance of metals in humid and oil medium environments [14,15].

Electro-explosive spraying is a modified technique in which partially molten metal droplets are sprayed on the substrate's surface at high speed, resulting in the formation of a single metallic layer [16–18]. This technique leads to the creation of a dense coating comprising fine and uniform particles in a short time. Metallurgical bonding potentiate provides various amorphous, microcrystalline, and nano-crystalline formations on various surfaces. Therefore, the physical and mechanical characteristics, such as wear and corrosion resistance and mechanical strength, could be enhanced [19–22].

According to the literature, Mo, W, and stainless steel could be successfully sprayed on aluminum and steel plates via electro-explosive spraying in the atmospheric environment [16,23,24]. Using the electro-thermal explosion spraying method based on plasma, Wei et al. [25] could coat an FeAl-based layer on a AISI1045 steel substrate. The obtained data presented that the FeAl coating layer comprising Cr and RE elements could result in finer coating, higher density, and superior oxidizing corrosion resistance in high temperatures. Romanov et al. [26] also used electro-explosive spraying to coat the ZnO-Ag mixture to enhance the wear and galvanic corrosion resistance of copper alloy surfaces. Huang et al. [16], in another attempt, utilized this technique to cover a AISI1045 steel substrate by a multilayer Mo/Cu/Fe composite, resulting in the formation of a sponge-like structure with a uniform thickness, high bonding strength, and an improved wear resistance.

In 2022, Wang et al. [18] also used this technique to single spray a Ag/graphite composite layer, presenting a proper adhesion of the coating layer with the applied substrate, appropriate metallurgical bonding interface, and deposition efficiency of 35%. Although several attempts have been deployed to evaluate the various outcomes of the electro-explosive spraying technique, there has been little discussion about evaluating the coated layer corrosion resistance. Therefore, in this study, a 321 austenitic stainless steel metal wire was applied as the spray material on the AISI1045 steel substrate because of the high corrosion resistance ability of 321 austenitic stainless steel. Then, the prepared sample was subjected to a simulated 168 h oil–water medium corrosion test. The influential primary parameters on the corrosion performance of electro-explosive spraying were investigated through examining the original structure, elements diffusion in the coating section, and corrosion products, as well as corrosion rate.

## 2. Experimental Procedures

### 2.1. Coating System Design

In this study, a commercially available 321 austenitic stainless steel wire alloy was sprayed on a AISI1045 steel substrate using an electro-explosive spraying device. The 321 austenitic stainless steel wire could provide a proper corrosion resistance behavior, due to its stabilization, by adding titanium against chromium carbide formation [27]. Therefore, this stabilized alloy is able to lead an excellent resistance to intergranular corrosion. As is described in Figure 1, the AISI1045 steel ($10 \times 10 \times 20$ mm$^3$) comprises C, Si, Mn, Cr, Ni, Cu, and Fe elements with the weight ratios of 0.45, 0.21, 0.60, 0.18, 0.25, 0.10, and balance wt.%, respectively. In addition, the 321 austenitic stainless steel (1.5 mm diameter) contains C, Si, Mn, Cr, Ni, S, P, and Fe with respective ratios of 0.08, 1.00, 2.00, 18.00, 11.00, 0.03, 0.045, and balance wt.%.

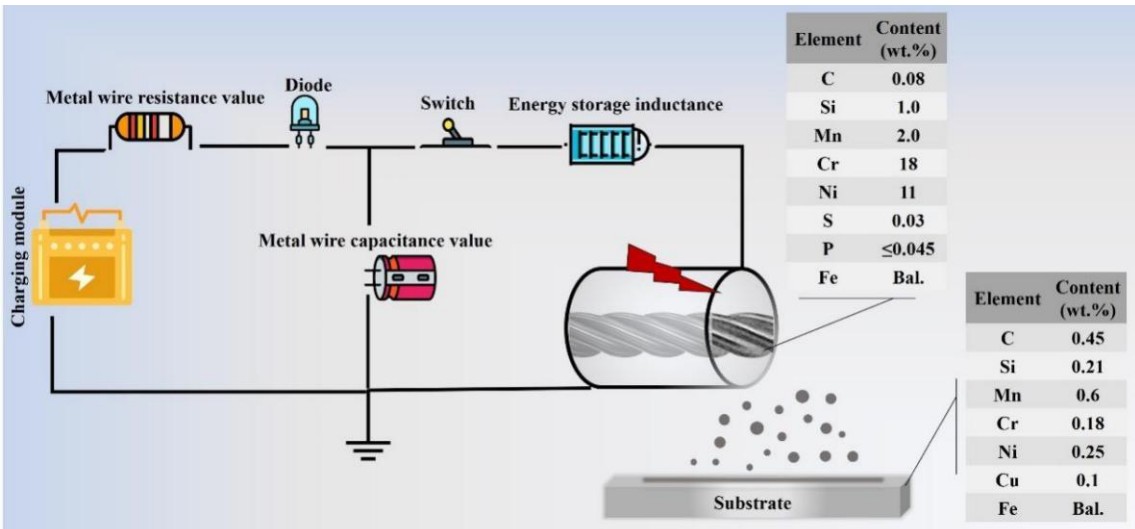

**Figure 1.** Schematic illustration of electro-explosive spraying experiment used for coating alloy 321 on AISI1045 steel.

A schematic illustration of the apparatus employed toward a successful coating is displayed in Figure 1. In the electro-explosive spraying technique, a high voltage is loaded on both ends of a metal wire to induce ohmic heating via an immediate high current. Then, the partially molten metal droplets are sprayed onto the substrate surface, generating a high-temperature- and ablation-resistant coating. In this equipment, the voltage was in the range of 0–20 kV, and the capacitor bank is composed of 28 capacitors, the capacity of each capacitor is 7.9 μF. Notably, the coating procedure was repeated 20 times [28].

### 2.2. Characterizations

Scanning electron microscopy (SEM, Sigma300, Zeiss, Germany), coupled with energy dispersive spectroscopy (EDS), was applied to evaluate the coating thickness and cross-sectional morphology of the coatings of the prepared specimens. Also, the elemental distribution of the coating was figured out using the X-ray diffraction patterns recorded in the scanning angle of $2\theta = 20°$ to $90°$ with the measuring rate of $2\theta/\text{min}$ (XRD, smartlab, Rigaku, Japan). The size of the crystalline planes was calculated using the Scherrer equation (Equation (1)) [29]:

$$\text{Dp} = (0.94 \times \lambda)/(\beta \times \cos\theta) \tag{1}$$

where Dp, β, θ, and λ are attributed to the average crystallite size, line broadening in radians, Bragg angle, and X-ray wavelength, respectively.

The bonding force between the coatings and the substrates was measured with a film FM1000 and a universal testing machine. The experiment was repeated four times, and the average values were reported. The electron backscattered diffraction (EBSD, JEOL, JSM-7900F, Tokyo, Japan) was also employed to analyze the coating grains. Furthermore, the coating hardness was measured with an automatic microhardness tester (Q10A+).

In order to examine the corrosion resistance, the surface of the sample is polished smooth with 600 # sandpaper. Then, the unsprayed surface was coated with epoxy resin, and the boundary was sealed with silica gel. Afterward, the samples were immersed in the corrosion solution for 168 h at a water bath temperature of 60 °C. The corrosion solution comprised calcium chloride ($CaCl_2$), sodium bicarbonate ($NaHCO_3$), sodium chloride (NaCl), sodium sulfate ($Na_2SO_4$), and Magnesium chloride contains 6 Water of crystallization($MgCl_2*6H_2O$). The concentration, ion species, and content of the highlighted components are summarized in Table 1. The samples were weighed, and the kinetic curve was generated every 24 h.

**Table 1.** Experimental medium composition of simulated oilfield (g/L).

| Compound | Concentration (mol/L) | Ion Species | Content (g/L) |
|---|---|---|---|
| $CaCl_2$ | 1.665 | $Ca^{2+}$ | 0.6 |
| $NaHCO_3$ | 0.826 | $HCO_3^-$ | 0.6 |
| NaCl | 30.715 | $Cl^-$ | 20 |
| $Na_2SO_4$ | 1.775 | $SO_4^{2-}$ | 1.2 |
| $MgCl_2 \cdot 6H_2O$ | 0.846 | $Mg^{2+}$ | 0.1 |

Then, the corrosion rate was calculated within 7 days using the following equation (Equation (2)):

$$R = \frac{0.76 \times 10^7 \times (m_0 - m_1)}{S \times T \times D} \tag{2}$$

where R is the corrosion rate (mm/a), $m_0$ is the mass before the experiment (g), $m_1$ is the mass after the experiment (g), S is the total area of the sample ($cm^2$), T is the experiment time (h), and D is the density of the material ($kg/m^3$).

## 3. Results

Electro-explosive spraying is a high-speed method, spraying droplets at high temperatures. High strength, as well as long-term durability, are the advantages declared for this process. So far, numerous studies have reported various outcomes and the dependent parameters of the electro-explosive spraying method. Meanwhile, there has been little discussion about the corrosion resistance of the coated film layer. As a result, we aimed to evaluate the corrosion resistance of the AISI1045 steel substrate covered with 321 metal wire droplets. Then, physiochemical characteristics, as well as the corrosion behavior of the provided specimen, were determined. The obtained results from the examinations are provided in the following sections.

### 3.1. Physiochemical Characteristics of the Coating Layer

The macroscopic and microscopic cross-sectional morphology of the coating film layer could easily reflect the coating quality. Figure 2a,b presents the cross-sectional morphologies of the provided layer on the 45 steal substrate in the macroscopic and microscopic views, respectively. Accordingly, the metal coating was densely and uniformly deposited on the surface of the substrate, with a thickness of about 137 µm. It is worth noting that no peeling sign was observed after subjecting the coated sample 50 times to thermal vibration experiments (heated to 350 °C for 30 min and then cooled to room temperature with the presence of water). Table 2 is the test results of bonding strength between coating and substrate. All samples in the experiment show detachment between the coatings and the substrates, and no detachment is found within the coatings. The average value of the measured bonding strength is about 53.239 MPa, as is shown in Table 2. The hardness of the original metal wire is 186 HV, and the microhardness of the coating after spraying is 232 HV.

**Table 2.** Bonding strength of the coatings.

| Coating | Bonding Strength (MPa) |
|---|---|
| Sample 1 | 53.825 |
| Sample 2 | 60.005 |
| Sample 3 | 47.369 |
| Sample 4 | 51.757 |
| Average | 53.239 |

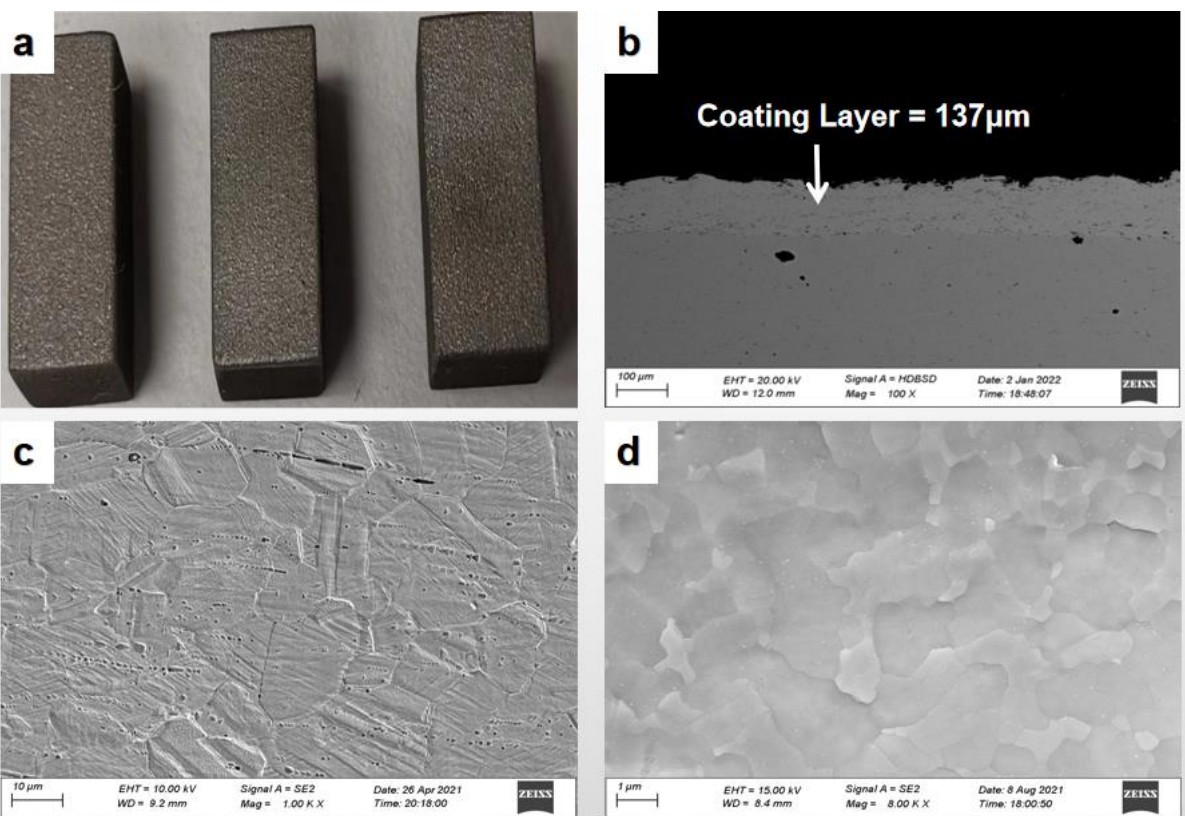

**Figure 2.** Morphological characteristics of the AISI1045 steel substrate coated with alloy 321 droplets; (**a**) macroscopic image of the provided specimen, (**b**) SEM image of the electro-explosive 321 austenitic stainless steel metal wire coating layer on the AISI1045 steel substrate, and SEM images of the 321 alloy wire (**c**) before and (**d**) after the coating.

The structure of 321 alloy wire before and after the coating procedure is figured out in Figure 3a,b. Based on the attained figures, the 321 austenitic stainless steel metal wire comprised grains in sizes ranging from 15 to 20 μm. However, the electro-explosive technique reduced the grain size to a range of 1–3 μm after the coating. Inhomogeneous white granular material was distributed in the coating periphery. Also, the microstructure retained the metallographic properties of austenitic steel after spraying. This could be assigned to the point that the 321 austenitic stainless steel is an austenitic stainless steel, which can re-form an austenitic structure throughout the high temperature and rapid cooling procedure, ensuring the coating's corrosion resistance.

The XRD pattern of the 321 austenitic stainless steel wire after the coating on the AISI1045 steel is depicted in Figure 3a. The appeared peaks in the 2θ degrees of 43.8, 50.94, and 74.87° are attributed to the Fe-Ni, while the ones in the range 2θ degrees of 44.69, 64.51, and 82.06° correspond to Fe-Cr bonds. Observing the characteristic peaks of the crystalline regions, the size of the crystalline planes was calculated using the Scherrer equation. Accordingly, the size of the crystalline planes was obtained as 27 (43.8°), 19 (44.69°), 19 (50.94°), 9 (64.51°), 16 (74.87°), and 9 nm (82.06°).

Figure 3b,c illustrates the TEM image and EDS spectrum of the coating surface on the AISI1045 steel substrate. The EDS spectrum represented peaks in the energy values of 0.81 (Fe), 0.85 (Ni), 0.69 (Cr), 0.58(O), 5.32 (Cr), 5.80 (Cr), 6.30 (Fe), 6.93 (Fe), 7.32 (Ni), and 8.11 keV (Ni), which corresponded to the presence of Fe, Ni, Cr, and O in the structure. Due to the alloy-based inherent structure of the 321 wire, O, Cr, Fe, and Ni elements appeared in the EDS spectrum with weight percentages of 3.21, 16.06, 70.67, and 10.06%. Based on the obtained data, the white particles present in the TEM image correspond to tiny oxides composed of Fe, Ni, and Cr elements with a size range of 30 to 50 nm. The

formation of the highlighted tiny Fe, Ni, and Cr oxides could be due to the formed oxidation product by the oxygen in the air, as well as the high-temperature metal droplets during the electric explosion.

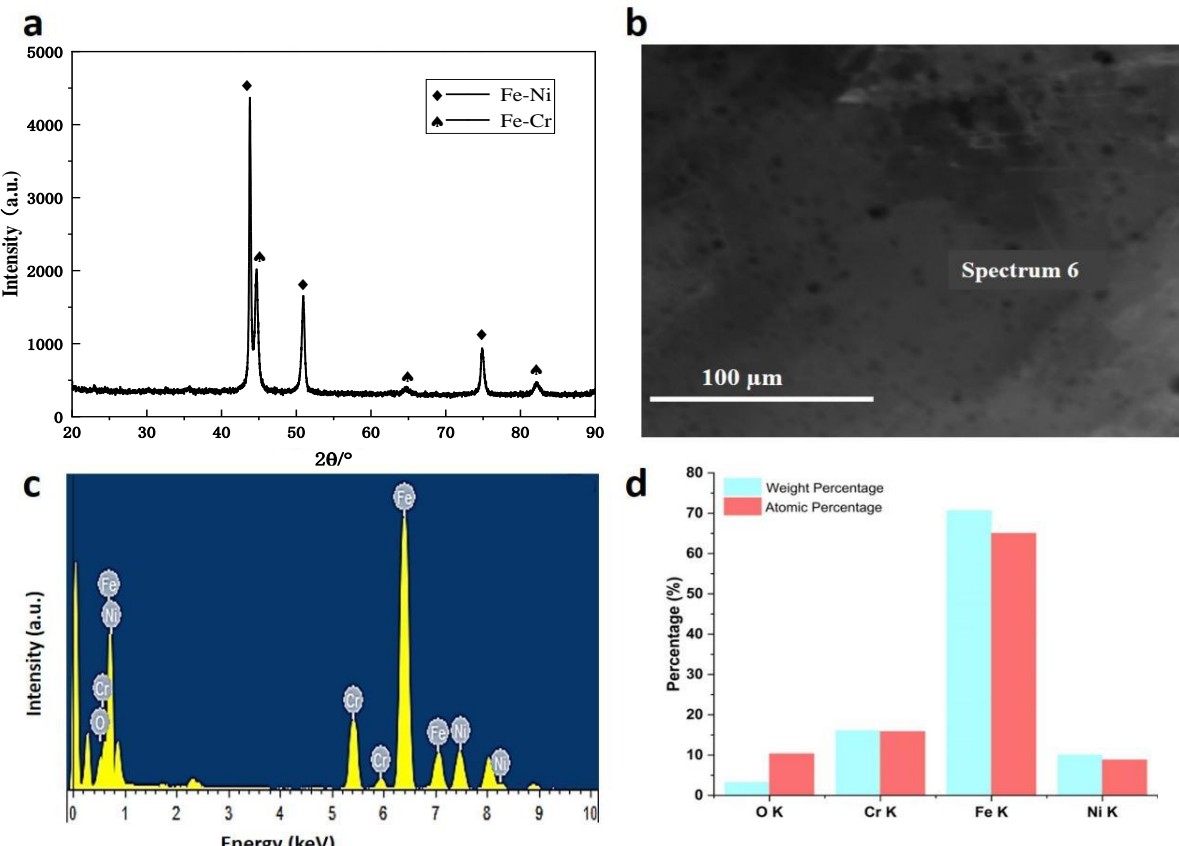

**Figure 3.** Characteristics of the employed 321 austenitic stainless steel for coating the AISI1045 steel; (**a**) XRD pattern, (**b**) TEM image, (**c**) EDS spectrum, and (**d**) the elemental ratio in the specimen.

The bonding strength results between the coating and the substrate are shown in Figure 4a. The experiment was carried out for three specimens, and the bonding strength values were measured. According to the obtained data, the approximate bonding strength of 49.21 MPa was estimated between the 321 austenitic stainless steel coating layer and the AISI1045 steel substrate.

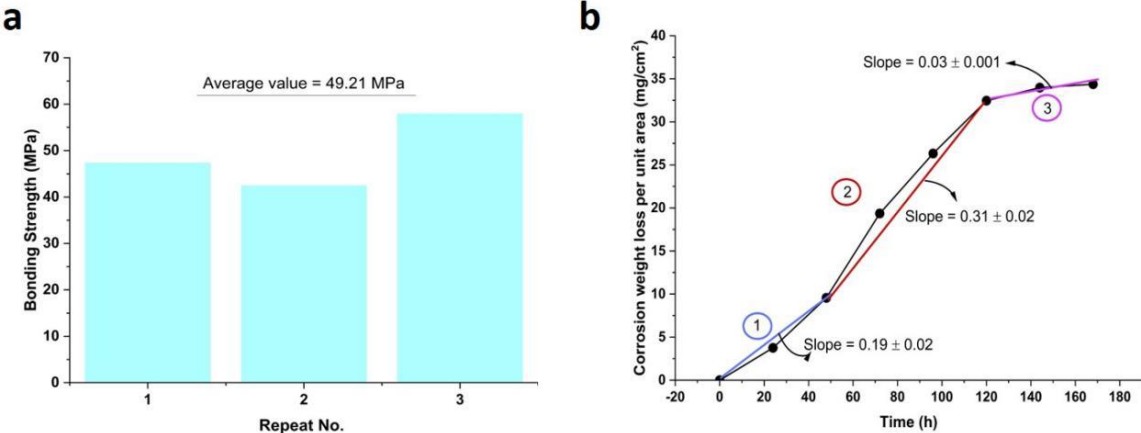

**Figure 4.** (**a**) Bonding strength between the alloy 321 and AISI1045 steel before the corrosion examination, and (**b**) corrosion kinetic curve of the coated sample after 168 h.

### 3.2. Corrosion Behavior of Coatings Prepared by Electric Explosion

The corrosion kinetic curve of the coating prepared by electro-explosion in the simulated oilfield medium, containing $CaCl_2$, $NaHCO_3$, NaCl, $Na_2SO_4$, and $MgCl_2 \cdot 6H_2O$ at 60 °C for 168 h, is displayed in Figure 4b. As is apparent, the total corrosion kinetics curve is parabolic. In addition, three regions were observed in the obtained kinetic curve with various slopes, including $0.19 \pm 0.02$, $0.31 \pm 0.02$, and $0.03 \pm 0.001$ mg/cm$^2$. In fact, the corrosion rate increased dramatically up to 40 h. Then, a gradual increase was observed up to 120 h, while it showed stable behavior after this time.

The XRD pattern of the coating surface prepared by electric explosion after corrosion at 60 °C for 168 h in oil and water is provided in Figure 5a. According to the pattern, the products after the corrosion procedure include $CaCO_3$, $Fe_3O_4$, and $MgFe_2O_4$. The Fe-Ni phase was observable in the spectrum, relating to the coating layer. Additionally, NaCl is the residue of the solute in the etching solution. It is worth noting that the Cr diffraction peak could be overlapped by NaCl peaks. Thus, it is not highlighted in the XRD pattern, while it could be present in the specimen. Figure 5b depicts a cross-sectional SEM image of the coated AISI1045 steel by 321 austenitic stainless steel droplets. The EDS spectra in two determined zones of the coated layer are also represented in Figure 5c,d. Based on the observed elements and their weight percentages, the presence of O and Cl elements were detected between the coating layer and the substrate. This might be related to the minute gaps in the coating, allowing corrosive ions in the solution to penetrate the coating and substrate, leading the coating and substrate to corrode [30].

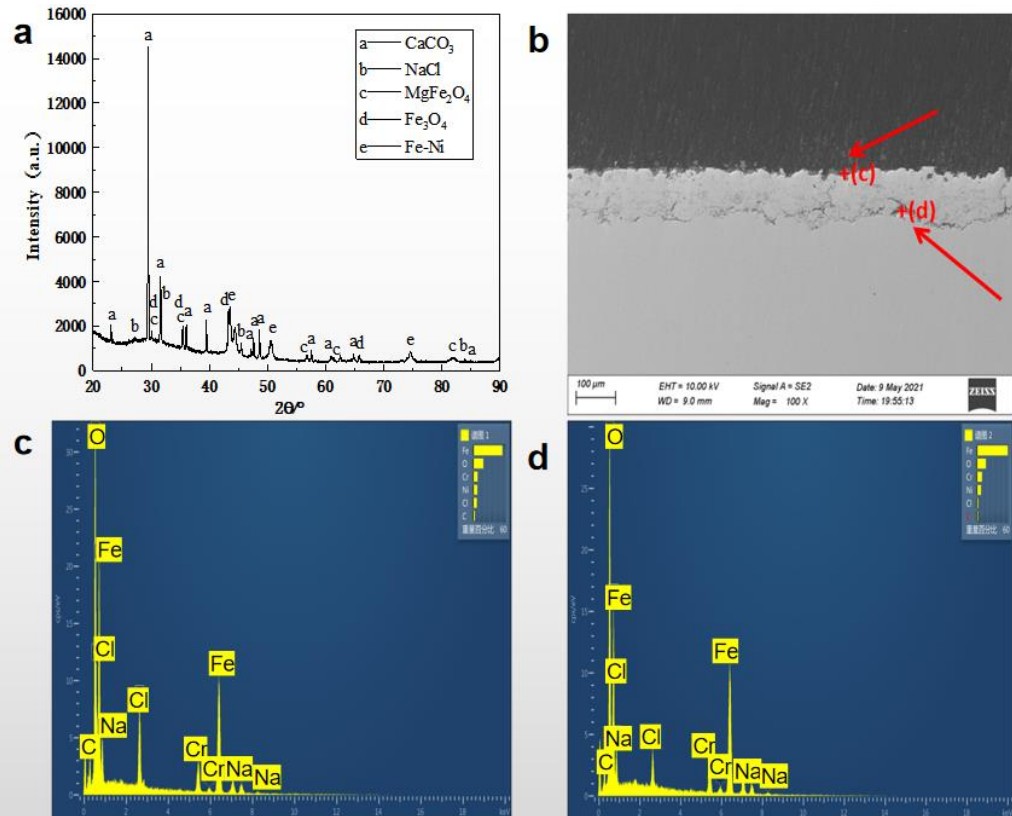

**Figure 5.** Morphological features of the specimen after corrosion in oil and water at 60 °C for 168 h: (**a**) XRD pattern, (**b**) SEM, and (**c,d**) EDS spectra in two specific zones.

Corrosion products of the coated samples were identified using an electron probe micro analyzer (EPMA), which is described in Figure 6. As could be seen in Figure 6a, the image mapping could be divided into three main regions, including substrate, bonding boundary, and the coating layer. Point analysis in various positions is also determined in

Figure 6b. Figure 6c–i also illustrates the distribution of different elements, including CP, Cr, C, O, Ni, Fe, and Ti in the coated sample after the corrosion examination. In the substrate, the presence of CP and Fe are highlighted, as is observable in Figure 6c,h. Meanwhile, Cr and Fe elements were distributed homogenously in the coating layer, which is effective in the enhancement of the corrosion resistance Figure 6d,h. Additionally, the presence of the O element was gradually decreased in the direction from the coating layer to the substrate. Moreover, Figure 6i confirms the presence of the Ti element with a low percentage in both coating and substrate layers.

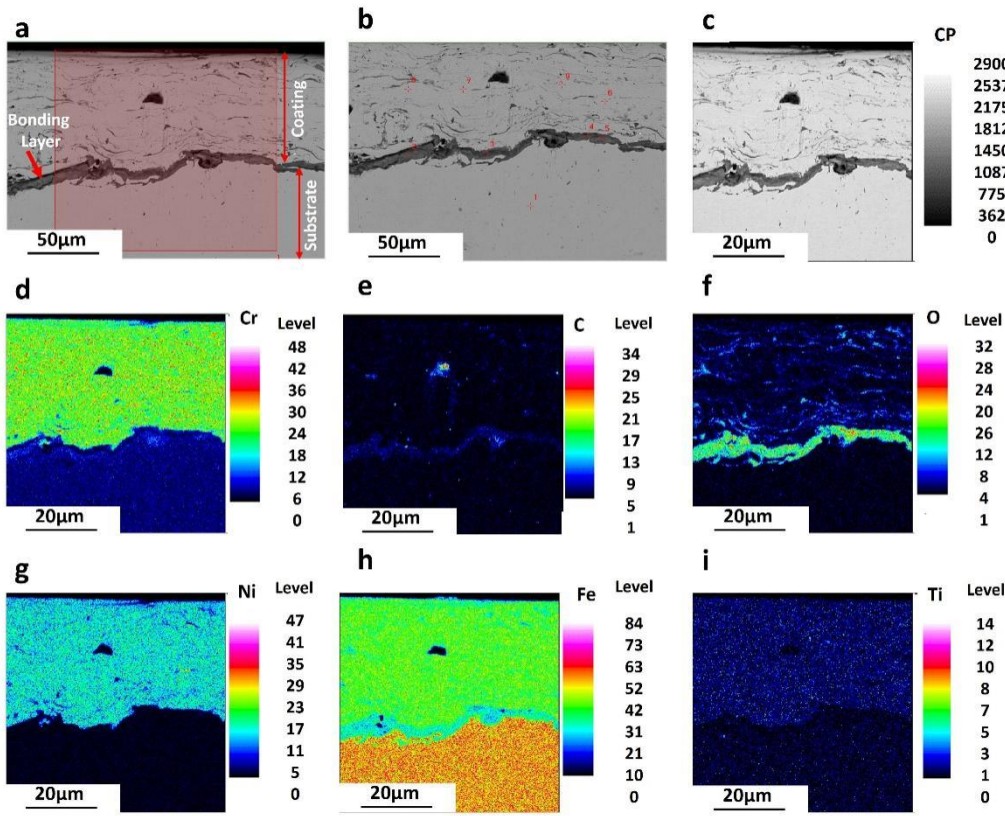

**Figure 6.** EPMA image mapping obtained after subjecting the coated AISI1045 steel by alloy 321 in an oilfield condition: (**a**) specific zone, (**b**) point analysis in different positions, and the contents of various elements and their distribution in the coated specimen, including (**c**) CP, (**d**) Cr, (**e**) C, (**f**) O, (**g**) Ni, (**h**) Fe, and (**i**) Ti.

## 4. Discussion

The electro-explosive spraying method influences sub-layer performances due to a short interaction time, small zone affected by the heat, and micron size of the coating particles [31]. In this study, we evaluated the morphological properties, as well as the corrosion resistance of the AISI1045 steel coated with 321 austenitic stainless steel metal wire through the electro-explosive spraying technique. Based on the results obtained from the SEM images, the grain size of the 321 austenitic stainless steel reduced from 15–50 μm to 1–3 μm after the coating procedure. The coating layer also showed a uniform surface with an average thickness of 137 μm. The EDS spectra confirmed the presence of Fe, Ni, Cr, and O elements in the coated sample, and the Fe-Ni, as well as the Fe-Cr bonds, were corroborated by the XRD pattern. In the XRD spectrum gained for the 321 austenitic stainless steel wire, the sharpest peak was obtained in the 2θ degree of 43.8°, belonging to austenitic phases, which ties well with the study conducted by Rezaei et al. [32]. The austenitic and martensitic are two major significant phases in the metal microstructures, which could be employed for recognizing the crystal structure at the atomic level of metals. The austenitic region could be easier welded compared to the martensitic phase. In addition,

the austenitic zone comprises less C content, thus causing superior corrosion resistance. Moreover, the chromium carbide would not be precipitated during the welding process in the austenitic phases. Furthermore, the austenitic phase has a cubic crystalline lattice in body center and is stable in high temperatures [33]. Therefore, the obtained data from the XRD spectrum revealed the desirable feature of the selected coating layer toward enhancing the corrosion behavior of the AISI1045 steel.

In the examination of the corrosion behavior in an oil and water condition at 60 °C, the primary corrosion reason could be attributed to the produced oxides during the explosion spraying process and microscopic pores inside the coating layer. First, the presence of oxides results in a potential difference with the surrounding non-oxides, leading to galvanic corrosion of the coating film layer. Second, the coating layer deploys some pores during the spraying process, causing the formation of micro galvanic corrosion in the pores and near the oxides. The chemical reactions in the following describe the combination of the aforementioned occurrences. In this process, the Fe element in the pores acts as an anode, while the dissolved oxygen and water molecules in the pores acts as a cathode (see Equations (3) and (4)).

$$\text{Anode}: Fe \rightarrow Fe^{2+} + 2e^- \tag{3}$$

$$\text{Cathode}: O_2 + 2H_2O + 4e^- \rightarrow 4OH^- \tag{4}$$

Therefore, $OH^-$ diffuses, and the pH at the orifice gradually increases, as shown in the following relation:

$$Fe^{2+} + 2OH^- \rightarrow Fe(OH)_2 \downarrow \tag{5}$$

$Fe(OH)_2$ is unstable and further reacts, leading to the formation of iron oxides as a result of the $Fe(OH)_3$ product and adhering to the coating surface in the presence of oxygen (Equation (6)).

$$4Fe(OH)_2 + 2H_2O + O_2 \rightarrow 4Fe(OH)_3 \downarrow \tag{6}$$

At the same time, with the increase of pH, the $HCO^{-3}$ ions in the corrosion medium gradually convert to $CO^{2-}{}_3$, which reacts with $Ca^{2+}$ in the corrosion solution to form $CaCO_3$ attached to the coating surface (Equation (7)).

$$CO_3^{2-} + Ca^{2+} \rightarrow CaCO_3 \downarrow \tag{7}$$

Simultaneously, $Mg^{2+}$ in the solution reacts with oxygen to generate MgO, which combines with $Fe_2O_3$ to form $MgFe_2O_4$. This type of corrosion formation is resulted from a mixture of chemical and electrochemical interactions (Equation (8)).

$$MgO + Fe_2O_3 \rightarrow MgFe_2O_4 \tag{8}$$

Therefore, the gradually generated $MgFe_2O_4$ oxide and spinel-structured corrosion product adhered to the coating's surface, which could be assigned to the presence of hydrated MgO in oil/aqueous environment. The $Mg^{2+}$ eases the electron transfer and sacrifices itself for the corrosion procedure [34]. Accordingly, a rather dense protective film, containing the deposited corrosion products, was generated, hindering the inward diffusion of oxygen elements and resulting in a drop in the corrosion rate after 120 h. Based on this, Figure 4b could be divided into three regions with different behaviors. The first area belonged to the corrosion trend before 40 h, in which the corrosion weight loss was increased with a slope of $0.19 \pm 0.02$. Then, the weight loss was gradually increased in the second area with a slope of $0.31 \pm 0.02$ up to 120 h. Afterward, the slope is reduced to $0.03 \pm 0.001$ due to the creation of corroded byproducts.

In addition, due to the presence of pores in the coating and $Cl^-$ ions in the solution, it can easily penetrate into the tiny gaps between the oxide films and form a channel between the substrate and the corrosive environment. This could lead to the enabling of the substrate material to exchange ions while the Fe element loses electrons. $Fe^{2+}$ is formed

in an oxygen-rich environment, progressively transforming into iron oxides. When these oxide layers are damaged at a specific place, the metal matrix and the undamaged part of the oxides make the surface passivated and form an activation–passivation corrosion battery. The passivated layer acts as a cathode that is considerably larger than the activation zone. Therefore, the corrosion develops deeply and creates a corrosion point between the coating and the substrate, primarily due to the reaction of the infiltrated corrosion elements with the iron in the substrate. The mentioned occurrences are well in agreement with the data obtained from the XRD diffraction pattern, SEM, and EDS, which are shown in Figure 5a–d. The XRD spectrum represented that the coated corrosion residue contained $MgFe_2O_4$, $Fe_3O_4$, and $CaCO_3$, leading to an improvement in corrosion resistance. This is due to the accumulation of $CaCO_3$ on the surface of corroded pores, causing slow ion exchange between the inside and outside, resulting in a decrease in the corrosion rate. The presence of Fe bonding with Ni and Cr elements was proved in the XRD spectrum of the alloy 321 in both before and after corrosion examination.

Compared to other analytical methods, EPMA provides satisfactory image mapping.

According to Figure 6, most of the oxides between the substrate and coating are formed of Fe elements. Based on the orientation of the oxides, it is possible to explain that corrosion is mainly caused by Fe elements, followed by Cr elements. The results indicated that the created oxides during the melting and solidification of the metal wire significantly influence corrosion. Additionally, the presence of the O element was gradually decreased in the direction of the coating layer towards the substrate, which could be due to the formation of a protective oxidation layer on the bonding boundary. Notably, a low concentration of Ti was corroborated in both substrate and coating layers, which can form a protective layer against corrosion because of a high affinity to oxygen [35]. Overall, the presence of highly concentrated coating layer as a primary component on the multi-phase corrosion layer directly influences the corrosion resistance and approaches stable behavior after a specific duration.

## 5. Conclusions

In conclusion, an AISI1045 steel substrate was covered with a protective layer containing 321 austenitic stainless steel molten metal particles using an electro-explosive spraying technique. The metallurgical characteristics, morphological features, and corrosion behavior of the produced specimen were studied. Accordingly, an external coating layer with a shrunk grain size of 1–3 μm and a uniform thickness of 137 μm was formed on the substrate. The bonding strength of 49.21 MPa was approached between the coating and substrate layers. Cr and Ni elements were highlighted in the XRD of the coated layer before subjecting the corrosion assay. The correlated data from the EDS and EPMA chemical analysis allowed us to evaluate the elemental features before and after the corrosion procedure. The corrosion examination illustrated three distinct regions and the maximum rate of 0.1961 mm/a at 120 h.

The main factor contributing to the corrosion of coatings prepared by electric explosion spraying of 321 austenitic stainless steel metal wires in oil and water is the enrichment of oxides at the grain boundaries. The enrichment of Cr elements at the grain boundaries leads to a decrease in Cr content within the grains, resulting in potential differences between the grain boundaries and the interior of the grains, leading to electrochemical corrosion.

Moreover, $CaCO_3$, $Fe_3O_4$, and $MgFe_2O_4$ were determined as the corrosion products by XRD analysis. The attained valuable results could be generated with the concomitant effects of Cr and Fe presence and their synergetic impacts with the Mg element. As a result, an outstanding protective layer on the AISI1045 steel could be formed by coating 321 austenitic stainless steel using a method that influentially preserves this substrate in oil–water conditions. Future studies are proposed to figure out the segregation and surface cracking of electro-explosive sprayed 321 austenitic stainless steel on the AISI1045 steel substrate, further controlling the enrichment of corrosion resistant elements at grain

boundaries, reducing the content of oxides inside the coating, and increasing the corrosion resistance of the coating.

**Author Contributions:** Conceptualization, Y.L. (Ye Liu). and Q.S.; methodology, Y.L. (Yali Liu); software, Y.L. (Ye Liu); validation, Y.L. (Ye Liu), H.D. and K.H.; formal analysis, Y.L. (Ye Liu); investigation, Y.L. (Ye Liu); resources, P.C.; data curation, Y.L. (Ye Liu); writing—original draft preparation, Y.L. (Ye Liu); writing—review and editing, Y.L. (Ye Liu); supervision, Q.S.; project administration, P.C. All authors have read and agreed to the published version of the manuscript.

**Funding:** This research was funded by the National Natural Science Foundation of China (Grant No. 11521062), State Key Laboratory of Explosion Science and Technology, Beijing Institute of Technology (Grant No. QNKT21-7).

**Data Availability Statement:** The data that support the findings of this study are available from the corresponding author upon reasonable request.

**Conflicts of Interest:** The authors declare no conflict of interest.

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
