# Peer review of "A Study on the Corrosion Resistance of a Coating Prepared by Electrical Explosion of 321 Metal Wire"

_lubricants, doi:10.3390/lubricants11070309_

Round 1

Reviewer 1 Report

lubricants-2496532

Study on corrosion resistance of coating prepared by electrical explosion of 321 metal wire

This is a good study. Authors have done a wonderful work which implies the beneficial role of electro-explosive sprayed alloy 321 in creating a protective layer against corrosion on 45 steel substrates in an enriched oil/water environment. Some of the major issues needs to be addressed before publication.

What made the authors to go for 4 samples only throughout the study.

Fig. 2: the magnification was different in b, c and d. Although Fig. 2b is different, the coating layer is not predominantly visible in the image. However, the magnification of Fig. 2c and Fig. 2d is completely different, it cannot be justified.

What is the black surface after the coated layer? Is the diffraction pattern or substrate?

How does the bonding boundary differ from the coating layer thickness?

What are concomitant effects between Fe and Cr with the impact of Mg element?

Some of the references needs to be updated with recent papers. The below references should be included in the revision.

https://doi.org/10.3390/ma16114182

https://doi.org/10.3139/120.111294

https://doi.org/10.1016/S1003-6326(22)65802-3

The conclusion should be rewritten with more data points and it should also explain about the future scope of this work.

I, as a reviewer of this manuscript, will accept this quality manuscript for publication after implementing all the minor corrections.

The linguistic level and the mechanics of English writing are not appropriate for publication. There are few grammatical and typing errors in the manuscript, so please check and revise. The way of writing is not clear and it is difficult for the readers to understand. The paper should be rewritten and proofread again thoroughly. Extensive editing of English language is required.

Author Response

Dear Reviewer,

Thank you very much for your time involved in reviewing the manuscript and your very encouraging comments on the merits.

We also appreciate your clear and detailed feedback and hope that the explanation has fully addressed all of your concerns. In the remainder of this letter, we discuss each of your comments individually along with our corresponding responses.

Comment 1:

What made the authors to go for 4 samples only throughout the study.

Response 1:

Thank you for the detailed review.I'm sorry, but my writing may not be detailed enough, which may have caused you to have a different understanding.In this experiment, we used 4 samples for force testing, 4 samples for corrosion testing, and 2 samples for TEM and cross-sectional testing.

Comment 2:

Fig. 2: the magnification was different in b, c and d. Although Fig. 2b is different, the coating layer is not predominantly visible in the image. However, the magnification of Fig. 2c and Fig. 2d is completely different, it cannot be justified.

Response 2:

Thank you for the detailed review.The reason for the inconsistency between the c and d scales in Figure 2 is that there is a significant difference in the grain size of the metal wire and the grain size after the formation of the coating. If modified to the same scale, it will result in a phenomenon where one image cannot display grain boundaries.We re edited Figure 2 and directly inserted the original data file. I hope to display the thickness and size of the coating more clearly.

Comment 3:

What is the black surface after the coated layer? Is the diffraction pattern or substrate?

Response 3:

Thank you for the detailed review.It is the inlay material used for the inlay coating. Because it is not conductive under SEM, it appears black in the figure.

Comment 4:

What are concomitant effects between Fe and Cr with the impact of Mg element?

Response 4:

Thank you for the detailed review.Magnesium chloride is corrosive in aqueous solution. When Magnesium chloride is dissolved in water, it will form strong acidic compounds such as hydrochloric acid (HCl) and magnesium hydroxide (Mg (OH) 2). These compounds react with metals and corrode their surfaces.Oxides enriched with Cr element at the grain boundary of the coating will be preferentially corroded. Oxides of Cr element have good corrosion resistance under dense conditions, but their corrosion performance is significantly reduced before a complete oxide film is formed. In acidic environments, Fe elements can also be corroded to form ferrous particles, which combine with OH in water to form Fe (OH) 2.

Comment 5:

Some of the references needs to be updated with recent papers. The below references should be included in the revision.

Response 5:

Thank you for the detailed review.I have carefully read the recommended article and inserted it into the references.

Comment 6:

The conclusion should be rewritten with more data points and it should also explain about the future scope of this work.

Response 6:

Thank you for the detailed review.The conclusion section has been supplemented.

Reviewer 2 Report

1)    The objective and novelty of the work is to be mentioned at the end of Introduction section.

2)    Please add the quantification of property enhancement for the composite in the ABSTRACT.

3)    Instead of designating as 45 steel, universally accepted designation system may be used as AISI1045 steel. Similarly, 321 steel may be named as 321 austenitic stainless steel.

4)    How coating parameters are selected? Please give atleast literature support.

5)    Please add literature support for equations 1 and 2.

6)    It is mentioned in section 2.2 as “In order to examine the corrosion resistance, the sample's surface was polished to a smooth finish”. What is the surface finish? Please quantify.

7)    Figures 2 c and d are of same scale. It is easy to compare if both are having the same scale. Please do accordingly.

8)    Figure 4a shows bonding strength between the coat and substrate. How the average value of 49.21 MPa is arrived? Why such large deviation is present between the trials? Please explain.

9)    In the Conclusions section, it is mentioned as “Future studies are proposed to figure out the segregation and surface cracking of electro-explosive sprayed alloy on the 45-steel substrate”. Where that proposal is given? Please elaborate.

10)                        It is highly recommended to format the Reference section.

Revision is required

Author Response

Dear Reviewer,

Thank you very much for your time involved in reviewing the manuscript and your very encouraging comments on the merits.

We also appreciate your clear and detailed feedback and hope that the explanation has fully addressed all of your concerns. In the remainder of this letter, we discuss each of your comments individually along with our corresponding responses.

Comment 1:

The objective and novelty of the work is to be mentioned at the end of Introduction section.

Response 1:

Thank you for the detailed review.The conclusion section has been supplemented and modified.

Comment 2:

Please add the quantification of property enhancement for the composite in the ABSTRACT.

Response 2:

Thank you for the detailed review.Quantified the microhardness of the coating before and after spraying.

Comment 3:

Instead of designating as 45 steel, universally accepted designation system may be used as AISI1045 steel. Similarly, 321 steel may be named as 321 austenitic stainless steel.

Response 3:

Thank you for the detailed review.Has been modified at the corresponding location.

Comment 4:

How coating parameters are selected? Please give atleast literature support.

Response 4:

Thank you for the detailed review.References have been added at the corresponding location.

Comment 5:

Please add literature support for equations 1 and 2.

Response 5:

Thank you for the detailed review.References have been added at the corresponding location.

Comment 6:

It is mentioned in section 2.2 as “In order to examine the corrosion resistance, the sample's surface was polished to a smooth finish”. What is the surface finish? Please quantify.

Response 6:

Thank you for the detailed review.The modifications have been made as required.

Comment 7:

Figures 2 c and d are of same scale. It is easy to compare if both are having the same scale. Please do accordingly.

Response 7:

Thank you for the detailed review.The reason for the inconsistency between the c and d scales in Figure 2 is that there is a significant difference in the grain size of the metal wire and the grain size after the formation of the coating. If modified to the same scale, it will result in a phenomenon where one image cannot display grain boundaries.We re edited Figure 2 and directly inserted the original data file. I hope to display the thickness and size of the coating more clearly.

Comment 8:

Figure 4a shows bonding strength between the coat and substrate. How the average value of 49.21 MPa is arrived? Why such large deviation is present between the trials? Please explain.

Response 8:

Thank you for the detailed review.There are many factors that affect the bonding strength of coatings, such as surface roughness, internal defects of coatings, etc.In addition, during the testing process, there may be some errors at the moment of separation between the coating and the substrate, which cannot be avoided during the experimental process.

Comment 9:

In the Conclusions section, it is mentioned as “Future studies are proposed to figure out the segregation and surface cracking of electro-explosive sprayed alloy on the 45-steel substrate”. Where that proposal is given? Please elaborate.

Response 9:

Thank you for the detailed review.Our team has been researching the preparation of coatings by electric explosive spraying since 2006, during which multiple materials have been tested. In previous experiments, we found that there would be partial segregation of alloy elements after explosion, so we chose 321 austenitic stainless steel because it contains Ti element, which can effectively suppress element segregation in the alloy.

Comment 10:

It is highly recommended to format the Reference section.

Response 10:

Thank you for your review. We will modify the format according to the journal's requirements.

Reviewer 3 Report

1-The introduction should provide more context and background information on different corrosion mechanism such as pitting, ion adsorbtion, and glavanic corrosion. In this regards following article must be added in intoduction.

https://doi.org/10.1016/j.corsci.2021.109485

https://doi.org/10.1016/j.diamond.2023.109793

https://doi.org/10.1016/j.corsci.2019.07.005

https://doi.org/10.1007/s12540-020-00692-y

2-It would be helpful to explain why the focus is on 45 steel substrate and 321 metal particulate fillers. What are their specific properties and relevance to the study?

3-The presence of Fe-Cr and Fe-Ni phases in the coating layer is mentioned, but their significance and impact on corrosion resistance should be discussed.

4-The corrosion rate of 0.1961mm/a up to 120h is provided, but it would be helpful to provide a comparison to other coatings or materials in similar conditions.

5-More information is needed on the corrosion products (CaCO3, Fe3O4, and MgFe2O4) and their effect on the overall corrosion resistance. How do they contribute to the protective properties of the coating?

6-It would be beneficial to provide any limitations or potential areas for improvement in the study.

Minor editing of English language required

Author Response

Dear Reviewer,

Thank you very much for your time involved in reviewing the manuscript and your very encouraging comments on the merits.

We also appreciate your clear and detailed feedback and hope that the explanation has fully addressed all of your concerns. In the remainder of this letter, we discuss each of your comments individually along with our corresponding responses.

Comment 1:

The introduction should provide more context and background information on different corrosion mechanism such as pitting, ion adsorbtion, and glavanic corrosion. In this regards following article must be added in intoduction.

Response 1:

Thank you for the detailed review.We have made modifications and cited references to the introduction section according to your suggestions.

Comment 2:

It would be helpful to explain why the focus is on 45 steel substrate and 321 metal particulate fillers. What are their specific properties and relevance to the study?

Response 2:

Thank you for the detailed review.Our team has been researching the preparation of coatings by electric explosive spraying since 2006, during which multiple materials have been tested. In previous experiments, we found that there would be partial segregation of alloy elements after explosion, so we chose 321 austenitic stainless steel because it contains Ti element, which can effectively suppress element segregation in the alloy.45 steel is only used as the substrate material in this article, providing a carrier for the coating.

Comment 3:

The presence of Fe-Cr and Fe-Ni phases in the coating layer is mentioned, but their significance and impact on corrosion resistance should be discussed.

Response 3:

Thank you for the detailed review.The presence of these two phases indicates that there was no significant phase transition before and after spraying, so the selected material can continue the corrosion resistance of stainless steel.

Comment 4:

The corrosion rate of 0.1961mm/a up to 120h is provided, but it would be helpful to provide a comparison to other coatings or materials in similar conditions.

Response 4:

Thank you for the detailed review.This article is a study on the corrosion of 321 austenitic stainless steel coating by electric explosion spraying. We will conduct tests in different spraying environments in the future. And compare the corrosion rate and further optimize the coating.

Comment 5:

More information is needed on the corrosion products (CaCO3, Fe3O4, and MgFe2O4) and their effect on the overall corrosion resistance. How do they contribute to the protective properties of the coating?

Response 5:

Thank you for the detailed review.According to your suggestion, explanations have been added to the analysis section.

Comment 6:

It would be beneficial to provide any limitations or potential areas for improvement in the study.

Response 6:

Thank you for your evaluation. We will make corresponding improvements in future research.
